# A Propensity Score Matched Analysis of Superparamagnetic Iron Oxide versus Radioisotope Sentinel Node Biopsy in Breast Cancer Patients after Neoadjuvant Chemotherapy

**DOI:** 10.3390/cancers14030676

**Published:** 2022-01-28

**Authors:** Zuzanna Pelc, Magdalena Skórzewska, Maria Kurylcio, Tomasz Nowikiewicz, Radosław Mlak, Katarzyna Sędłak, Katarzyna Gęca, Karol Rawicz-Pruszyński, Wojciech Zegarski, Wojciech P. Polkowski, Andrzej Kurylcio

**Affiliations:** 1Department of Surgical Oncology, Medical University of Lublin, Radziwiłłowska 13 St., 20-080 Lublin, Poland; magdalena.skorzewska@umlub.pl (M.S.); m.kurylcio@gmail.com (M.K.); sedlak.katarz@gmail.com (K.S.); kasiaa.geca@gmail.com (K.G.); karolrawiczpruszynski@uml.edu.pl (K.R.-P.); wojciechpolkowski@umlub.pl (W.P.P.); andrzej.kurylcio@umlub.pl (A.K.); 2Department of Clinical Breast Cancer and Reconstructive Surgery, Oncology Center, Prof. Lukaszczyk Memorial Hospital, Romanowskiej 2 St., 85-796 Bydgoszcz, Poland; tomasz.nowikiewicz@cm.umk.pl (T.N.); zegarskiw@cm.umk.pl (W.Z.); 3Department of Human Physiology, Medical University of Lublin, Radziwiłłowska 11 St., 20-080 Lublin, Poland; radoslaw.mlak@gmail.com

**Keywords:** breast cancer, neoadjuvant chemotherapy, sentinel lymph node biopsy, superparamagnetic iron oxide, SPIO, radioisotope

## Abstract

**Simple Summary:**

This Propensity Score Matched Analysis aimed to assess the efficacy of superparamagnetic iron oxide (SPIO) and radioisotope sentinel lymph node biopsy (SNLB) in breast cancer (BC) patients after neoadjuvant chemotherapy (NAC). One hundred and twenty-four patients were eligible for final analysis. In the SPIO group, the median of retrieved sentinel lymph nodes (SLNs) was significantly higher than in the RI group. The SPIO method was associated with a significantly higher chance of retrieving at least three SLNs when compared to the RI method. SPIO-guided SLNB allows efficient retrieval and detection of SLNs in BC patients after NAC when compared to RI.

**Abstract:**

The standard method for nodal staging in breast cancer (BC) patients after neoadjuvant chemotherapy (NAC) is sentinel lymph node biopsy (SLNB) with a radioisotope (RI) injection. However, SLNB after NAC results in high false-negative rates (FNR), and the RI method is restricted by nuclear medicine unit dependency. These limitations resulted in the development of the superparamagnetic iron oxide (SPIO) method, reducing FNR and presenting a comparable detection rate. This bi-institutional cohort comparison study aimed to assess the efficacy of SPIO and radioisotope SNLB in BC patients after NAC using Propensity Score Matching (PSM) analysis. The study group comprised 508 patients who underwent SLNB after NAC for ycT1-4N0M0 BC between 2013 and 2021 in two high volume centers. Data were retrieved from prospectively conducted databases. In the SPIO group, the median of retrieved sentinel lymph nodes (SLNs) was significantly higher than in the RI group (3 vs. 2; *p* < 0.0001). The SPIO method was associated with a significantly higher chance of retrieving at least three lymph nodes when compared to the RI method (71% vs. 11.3%; *p* < 0.0001). None of the analyzed demographic and clinical variables had a statistically significant influence on the efficacy of SLNs retrieval in the RI group, while in the SPIO group, patients with ≥three harvested SLNs had lower weight and decreased BMI. Based on this PSM analysis, SPIO-guided SLNB allowed the efficient retrieval and detection of SLNs in BC patients after NAC compared to RI.

## 1. Introduction

Sentinel lymph node biopsy (SLNB) is an established procedure for clinically negative nodes (cN0) in early breast cancer (BC) [1,2]. Since the end of the 20th century, the introduction of the SLNB developed a minimally invasive staging procedure for BC patients [3]. However, the surgical management of the axilla has been a matter of debate. Neoadjuvant chemotherapy (NAC) combined with personalized, targeted therapy results in a high pathologic complete response in the primary tumor [2]. Moreover, SLNB performed after NAC decreases the axillary lymph node dissection (ALND) rate, reducing adverse effects such as seroma, wound infections, or haemorrhage [4,5]. One of the factors restricting proper axillary mapping after neoadjuvant therapy includes the alteration of the lymphatic drainage due to fibrosis and obstruction of lymphatic vessels or the apoptosis of tumor cells [6]. For this reason, SLNB after NAC results in false-negative rates (FNR) varying from 10 to 30%, as shown in the SENTINA trial [7]. The current standard for nodal staging in BC patients after NAC is a radioisotope (RI) SLNB. However, this method also contains disadvantages such as nuclear medicine unit dependency or radiation exposure [8]. The existing drawbacks resulted in new, non-radioactive methods of sentinel lymph node (SLN) identification. In a recent meta-analysis, fluorescence-guided SLNB using indocyanine green (ICG) occurred to present non-inferior IR to the current standard [9]. In the randomized study, Jung et al. compared SLNB with ICG versus dual technique for BC patients after NAC, proving ICG to be a feasible method, with no statistically significant higher IR than RI-alone [10]. Superparamagnetic iron oxide (SPIO) with a handheld magnetometer presents a comparable detection rate as the RI combined with blue dye, known as the dual technique [11]. Moreover, the SPIO tracer reduces the FNR, possibly due to its peculiar, one-dimensional nanostructure and various physicochemical properties dependent on high intrinsic anisotropy and surface activity [12,13,14]. However, Corso et al. suggest a significant discrepancy between the FNR and detection rate depends on the individual experience of the research center or incoherent structure of retrospective studies [15]. Numerous trials and meta-analyses revealed the noninferiority of the SPIO to the gold standard isotope technique [16,17,18,19,20,21,22,23,24,25,26,27]. However, these studies did not include BC patients after NAC. Recently, we have determined that SPIO is a feasible and oncologically safe method for identifying SN in BC patients after preoperative treatment [17]. Therefore, the present study aimed to compare the identification rate (IR) of SPIO-guided SLNB to RI in BC patients after receiving neoadjuvant treatment based on the Propensity Score Matching (PSM) analysis.

## 2. Materials and Methods

### 2.1. Study Design

This bi-institutional cohort comparison study was performed based on the data retrieved from prospectively conducted databases. The study group comprised 508 patients who underwent SLNB after NAC for non-recurrent, non-metastatic ycT1-4N0M0 BC between 2013 and 2021 in two high volumes centers. After PSM analysis, 124 patients were eligible for analysis. Institutional review board approval (Bioethical Committee of the Medical University of Lublin, Ethic Code: Ke-0221-34-2013) was granted. PSM was performed to eliminate selection bias of clinicopathological data of enrolled patients (Figure 1).

### 2.2. Neoadjuvant Chemotherapy

Each patient was qualified for neoadjuvant treatment based on the multidisciplinary team decision. NAC was administered in accordance with national guidelines, depending on the clinical stage of the disease and molecular subtype. Four cycles of conventional or dose-dense AC (doxorubicin 75 mg*/*m^2^ with cyclophosphamide 750 mg*/*m^2^) followed by 12 cycles of weekly paclitaxel (80 mg*/*m^2^) or triweekly docetaxel (100 mg*/*m^2^) were the preferred regimen. Patients with primary BC with human epidermal growth factor receptor, two (HER2) protein overexpression, and/or HER2 gene amplification (HER2–positive) additionally received anti-HER2 therapy. The evaluation of the pathological tumor response to the NAC was performed according to Pinder classification [23]: response—complete pathologic response or <10% residual tumor/tumor bed; no response—≥10% residual tumor/tumor bed.

### 2.3. Sentinel Lymph Node Biopsy

In the RI method, the isotope 99mTc with an activity of 75–100 MBq was used and administered on an albumin carrier (Nanocol). About 2–3 h before the surgery, lymphoscintigraphy was performed. The radiotracer was administered as a periareolar intradermal injection into the lesion-specific breast quadrant. We used a manual gamma radiation detector (Crystal Prob GmBH, Berlin, Germany; Crystal Photonics GmbH, Berlin, Germany; NeoProbe, San Diego, California; Autosuture, Cornwalk, Conn) to perform intraoperative identification of areas of increased radiotracer capture within the axilla and measurements of radiation levels (hot spot). 

A handheld magnetometer (SentiMag^®^, Sysmex Europe GmBH, Hamburg, Germany) was used in the SPIO method. This allowed for a non-radioactive detection and location of SLNs before their surgical retrieval. Sienna+^®^ (Sysmex Europe GmBH, Hamburg, Germany) and Magtrace^®^ (Endomagnetics Limited, Cambridge, UK) were used. SPIO was injected deeply into the subareolar interstitial tissue, followed by SentiMag^®^ probe measurements as previously described [17].

All visualized SLNs were removed regardless of the method until the background signal was less than 10% of its highest value during SLNB. The method was assessed as efficient when retrieving at least three SLNs, confirmed in detailed histopathological reports. These assumptions are in accordance with the 2019 St. Galen Consensus Conference [18]. For the study purposes, the following definitions were adopted: SLN retrieval—intraoperative assessment and removal of at least three SLNs for further pathological analysis; SLN evaluation—the objective histopathological verification of separately submitted SLNs specimens. 

### 2.4. Statistical Analysis

Data were analyzed using MedCalc Statistical Software version 20.009 (MedCalc Software, Ostend, Belgium). Due to the two-institutional character of the study, Propensity Score Matching (PSM) was used, enabling matching patients by their clinicopathological features before further analysis. Initially, the SPIO group counted 74 patients, whereas the RI group had 434 patients. After PSM, considering age, ypTN, and biological subtype, each group consisted of 62 patients. Due to the non-normal distribution of continuous data (assessed using the D’Agostino-Pearson test), non-parametric tests were used (Mann U–Whitney test) to compare demographic and clinical variables between SPIO and RI groups). The chi-square test was used to compare the distribution of individual variants of categorical variables in both studied groups. The odds ratio (OR) test was used to assess the chance of achieving the desired efficacy of a given method of SLNs detection/retrieval (evaluated by surgeon or pathologist) (SPIO vs. RI). In all cases, *p*-value < 0.05 was considered statistically significant. The results for which *p* had values were >0.05, and <0.06 was considered a trend towards significance.

## 3. Results

### 3.1. Characteristics and Comparison of the Study Groups

#### 3.1.1. RI

The RI group consisted of 62 women with a median age of 52. The median BMI was 25.72. The overweight or obese women dominated (56.5%). The dominant clinical features were as follows: ypT0 (56.5%) and ypN0 (91.9%). The most common biological types were B1 (33.9%), TN (29%), B2 (22.6%), HER2 + (11.3%), and A (3.2%), respectively. Most patients had pathological tumor responses to NAC (69.4%). Almost half of the patients underwent breast conserving surgery (BCS) (48.4%). ALND was performed in all patients with lymph nodes metastases (9.7%). 

#### 3.1.2. SPIO

The SPIO group consisted of 62 women with a median age of 53.5 years. The median BMI was 25.97. Overweight or obese patients dominated (61.3%). ypT0 (56.5%), and ypN0 (91.9%) were dominant clinical features. The distribution of biological types in this group was identical to that in the RI group. Fifty-eight point one percent (58.1%) of the patients presented pathological tumor responses to NAC. Fifty-one point six percent (51.6%) of the patients underwent BCS. ALND was performed in all cases with lymph nodes metastases (8.1%). Detailed characteristics and comparisons of the study groups are presented in Table 1.

### 3.2. Comparison of SLNs Detection Efficacy Depending on the Method Implemented (RI vs. SPIO)

In the SPIO group, the median of retrieved SLNs was significantly higher than in the RI group (3 vs. 2, *p* < 0.0001). Similarly, the median of evaluated SLNs in the SPIO group was significantly higher than in the RI group (four vs. three, *p* = 0.0005). The SPIO method was associated with a significant, over 19-fold higher chance of retrieving at least three SLNs when compared to the RI method (71% vs. 11.3%; OR = 19.21; 95% CI: 7.36–50.10; *p* < 0.0001). Moreover, the SPIO method was associated with a significant, over 3-fold higher chance of evaluation of at least three SLNs when compared to the RI method (71.4% vs. 51.6%; OR = 3.21; 95% CI: 1.48–6.98; *p* = 0.0032). Detailed comparisons of the effectiveness of SLNs retrieval in both groups are shown in Table 2.

In the SPIO group, patients with SLN harvest ≥ 3 had lower weight (median: 66 vs. 77 kg; *p* = 0.0280) and lower BMI (25.55 vs. 28.26 kg/m^2^; *p* = 0.0323). None of the analyzed demographic and clinical variables had a statistically significant influence on the efficacy of SLNs retrieval in the RI group. Detailed data on the influence of selected demographic and clinical variables on the efficacy of SLNs retrieval in the RI and SPIO groups are presented in Table 3.

None of the analyzed demographic and clinical variables had a statistically significant influence on the efficacy of SLNs evaluation in the RI and SPIO groups. Detailed data on the influence of selected demographic and clinical variables on the efficacy of SLNs evaluation in the RI and SPIO groups are shown in Appendix A.

## 4. Discussion

To the best of our knowledge, apart from the recent IMAGINE study, this is the first research that compared RI versus SPIO for SLNB after NAC in BC patients [28]. In both studies, the SPIO method was compared to the RI-alone. 

Our results indicate that retrieved and evaluated SLNs’ median was significantly higher in the SPIO group. Higher levels of detected SLNs in the magnetic tracer group were observed in the present and IMAGINE studies. However, the median number of detected SLNs in the latter study ranged from 1.3 to 1.4. In order to perform a proper evaluation of SLNB after systemic treatment and decrease the risk of FNR, we aimed to assess at least three SLNs [20,21,29,30]. As shown in ACOSOG Z1071 trial, SLN IR and FNR were approximate between SLNB before and after NAC in patients with cN0 [29]. Notably, the sensitivity of the assessment increased with the retrieval of two or more SLNs, supporting the recommendations to perform SLNB in the post-neoadjuvant setting. 

Since AMAROS’ trial results, the gold standard for SLNB in BC patients remains the RI technique, which presents a comparable detection rate to the dual technique [12,19,31]. Numerous studies confirmed the noninferiority of SPIO to RI in the upfront surgery setting [16,17,24,25,32]. Since 2014, when Rubio et al. presented promising data on the outcome of SLNB after NAC using a dual tracer (SPIO-TC99) at the ASCO annual meeting [33], the results of the SENTINAC-01 clinical trial are awaited [34]. The study primary and secondary outcomes measures FNR and detection rate, respectively. After neoadjuvant treatment, BC patients are randomized into three groups, with SLNB guided by dual technique, combined RI + SPIO or SPIO alone. Our previous study established SPIO-guided SLNB after NAC as a safe and feasible method [17].

Surprisingly, Aksoy et al. indicated that SLNB after NAC did not influence overall survival or disease-free survival, underlining the necessity of adequate NAC response assessment [35]. However, the evaluation of SLNs after NAC provides more precise predictions of patients’ response to systemic therapy and residual disease severity [36], indicating that SLNB remains an essential tool for evaluating systemic treatment [36]. Jatoi et al. concluded that SLNB after NAC presented disadvantages such as decreased IR and increased FNR due to the lymphatic drainage alteration [37]. However, in the SPIO group of the present study, IR and FNR were 98.4% and 0%, respectively. These results support the hypothesis that SPIO seems to be an optimal tracer for SLNB after NAC in BC patients due to high SLN retrieval number and low FNR compared with the conventional methods [22]. This conclusion was confirmed by Mok et al., suggesting SPIO-guided SLBN improves the clinical value of SLNB with similar accuracy but avoids irradiation or risks of allergy to blue dye [8,23]. Furthermore, SLNBs using the Resovist magnetic nanoparticles and a new handheld, lightweight magnetic probe in BC patients were considered equivalent to the conventional RI method, as shown in a recent multicenter Japanese trial [24].

There are several limitations associated with the use of magnetic-guided surgery, including limited depth of detection and restricted use of metal instruments [38]. Moreover, since the magnetometer must seek out a small tracer collection point, the detection of SLNs by SPIO may be limited. In contrast, the RI probe can detect the radiation beam directly from SLNs [39]. Peek et al. investigated injection characteristics of the magnetic tracer on the rat model, proving that iron uptake appears to be proportional to the injection dose before reaching a plateau level [40]. This outcome suggests that each lymph node presents maximum magnetic particles load depending on the size and location of the SLN. In order to increase the iron uptake, the injection preferably should be performed one day before the SLNB, rather than directly prior to the surgery.

The RI method is also associated with specific limitations such as short half-life limiting the timeframe of SLNB and dependency on the availability of nuclear medicine units or disturbed SLN detection. Additionally, the magnetic tracer is related to no radiation exposure, easier implementation, longer retention in the SLNs, and more convenient workflow than the dual technique [30]. Although SPIO-guided SLNB seems to be an equivalent method to RI, the standardization of the axillary nodal management in BC patients after NAC is warranted.

The presented study has some limitations. Despite PSM analysis, the associated selection bias may exist. The databases were conducted since 2013 when the guidelines did not specify the necessity of retrieval and detection of at least three SLNs during SLNB.

## 5. Conclusions

In patients with BC after NAC, SLNB using magnetic technique allows high IR of SLNs and may result in more efficient retrieval and detection of at least three SLNs compared to RI. Considering the increasing role of preoperative chemotherapy and noninferiority of SPIO to the current standard, further studies establishing ferromagnetic assessment of SLNB after NAC are indicated.

## Figures and Tables

**Figure 1 cancers-14-00676-f001:**
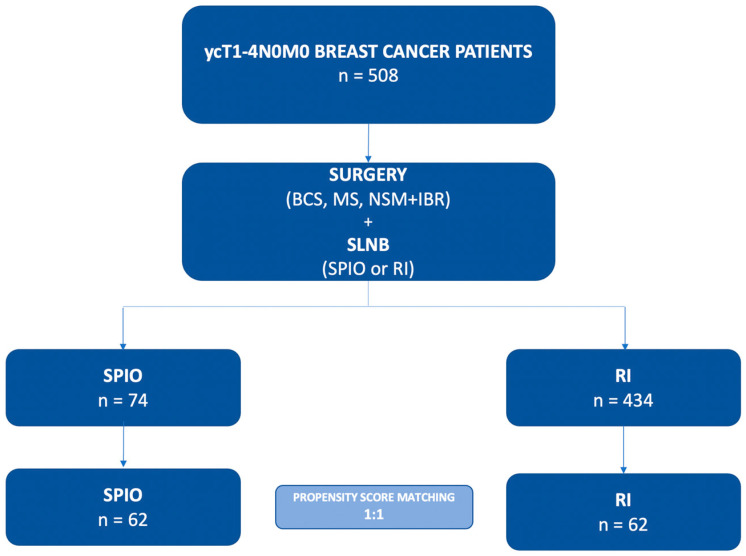
Flow chart of the Study Group. BCS—breast conserving surgery; MS—simple mastectomy, NSM—nipple-sparing mastectomy; IBR—immediate breast reconstruction; SLNB—sentinel lymph node biopsy; SPIO—superparamagnetic iron oxide; RI—radioisotope.

**Table 1 cancers-14-00676-t001:** Characteristics and comparison of the study groups.

Variable	RI	SPIO	*p*
	(*n* = 62)	(*n* = 62)	
Age (years)			
Median (interquartile range)	52 (44–61)	53.5 (43–62)	0.7910
Weight (kg)			
Median (interquartile range)	70 (62–75)	68 (58–77)	0.7413
BMI			
Median (interquartile range)	25.72 (23.31–29.33)	25.97 (21.60–28.63)	0.6601
BMI			0.3933
Underweight	2 (3.2%)	1 (1.6%)	
Healthy body weight	25 (40.3%)	23 (37.1%)	
Overweight	21 (33.9%)	27 (43.5%)	
Obese (Grade I)	10 (16.1%)	8 (12.9%)	
Obese (Grade II)	-	2 (3.2%)	
Obese (Grade III)	4 (6.5%)	1 (1.6%)	
Tumor highest diameter (mm)	19.5 (15–30)	25 (15–30)	0.4044
ypT			0.9760
0	36 (58.1%)	35 (56.5%)	
1	14 (22.6%)	15 (24.2%)	
2	12 (19.4%)	12 (19.4%)	
ypN			0.7415
Negative	57 (91.9%)	57 (91.9%)	
Positive	5 (8.1%)	5 (8.1%)	
ypTN			1.0000
T0N0	36 (58.1%)	36 (58.1%)	
T1N0	11 (17.7%)	11 (17.7%)	
T1N1	3 (4.8%)	3 (4.8%)	
T2N0	10 (16.1%)	10 (16.1%)	
T2N1	2 (3.2%)	2 (3.2%)	
Biological subtypes of cancer			1.0000
A	2 (3.2%)	2 (3.2%)	
B1	21 (33.9%)	21 (33.9%)	
B2	14 (22.6%)	14 (22.6%)	
HER2+	7 (11.3%)	7 (11.3%)	
TN	18 (29%)	18 (29%)	
NAC	62 (100%)	62 (100%)	1.0000
Response to NAC			0.2625
No response (small, medium)	19 (30.6%)	26 (41.9%)	
Response (high, complete)	43 (69.4%)	36 (58.1%)	
Type of surgery			0.1419
BCS	32 (51.6)	30 (48.4%)	
MRM	3 (4.8%)	1 (1.6%)	
MS	23 (37.1%)	19 (30.6%)	
NSM+IBR	4 (6.5%)	12 (19.4%)	
Site			1.0000
Left	34 (54.8%)	33 (53.2%)	
Right	28 (45.2%)	29 (46.8%)	
Margin			0.1273
R0	62 (100%)	58 (93.5%)	
R1	-	4 (6.5%)	
Lymphadenectomy			1.0000
No	56 (90.3%)	57 (91.9%)	
Yes	6 (9.7%)	5 (8.1%)	
ycSNB (retrieved)			<0.0001 *
Median (interquartile range)	2 (2–2)	3 (2–4)	
ypSNB (evaluated)			0.0005 *
Median (interquartile range)	3 (2–3)	4 (3–5)	
ypSN			0.7415
Negative	57 (91.9%)	57 (91.9%)	
Positive	5 (8.1%)	5 (8.1%)	

BMI—body mass index; RI—radioisotope; SPIO—superparamagnetic iron oxide; ypT—post neoadjuvant therapy T stage; ypN—post neoadjuvant therapy N stage; ypTN—post neoadjuvant therapy T and N stage; A—luminal A; B1– luminal B HER2 negative; B2—luminal B HER2 positive; HER2+—human epithelial receptor-positive; TN—triple-negative; NAC—neoadjuvant chemotherapy; BCS—breast conserving surgery; MRM—modified radical mastectomy; MS—simple mastectomy; NSM—nipple-sparing mastectomy; IBR—immediate breast reconstruction; R0—radical microscopic margin; R1—non-radical microscopic margin; ycSNB—post neoadjuvant therapy clinical sentinel lymph node biopsy number; ypSNB—post neoadjuvant therapy pathological sentinel lymph node biopsy number; ypSN—post neoadjuvant therapy pathological sentinel lymph node number; *—statistically significant result.

**Table 2 cancers-14-00676-t002:** Comparison of LN’s retrieval efficacy in RI and SPIO methods.

Variable	RI	SPIO	OR [95%CI]
	(*n* = 62)	(*n* = 62)	*p*
SLN retrieval			
<3 retrieved SLNs	55 (88.7%)	18 (29%)	19.21 [7.36–50.10]
≥3 retrieved SLNs	7 (11.3%)	44 (71%)	<0.0001 *
SLN evaluation			
<3 evaluated SLNs	30 (48.4%)	14 (22.6%)	3.21 [1.48–6.98]
≥3 evaluated SLNs	32 (51.6%)	48 (77.4%)	0.0032 *
Efficacy of positive SLNs detection			
Negative	57 (91.9%)	57 (91.9%)	1.00 [0.30–3.28]
Positive	5 (8.1%)	5 (8.1%)	1.0000
IR of SLNs retrieval			
Undetected SLNs	-	-	0.33 [0.01–8.21]
Detected SLNs	62 (100%)	62 (100%)	0.4974
IR of SLNs evaluation			
Undetected SLNs	-	-	0.19 [0.01–4.42]
Detected SLNs	62 (100%)	62 (100%)	0.2924

SLNs—sentinel lymph nodes; RI—radioisotope; SPIO—superparamagnetic iron oxide; IR—identification rate; OR—odds ratio; *—statistically significant result.

**Table 3 cancers-14-00676-t003:** Influence of selected demographic and clinical variables on the efficacy of SLNs detection/retrieval (based on surgeon evaluation) using the RI or SPIO method.

Variable	RI (*n* = 62)		*p ^a^*	SPIO (*n* = 62)		*p ^a^*
			or			or
	SLNs Retrieval Efficacy		OR (95%CI)	SLNs Retrieval Efficacy		OR (95%CI)
	<3 Retrieved SLNs	≥3 Retrieved SLNs	*p ^b^*	*p ^b^*	≥3 Retrieved SLNs	*p ^b^*
Age (years)						
Median (interquartile range)	53 (44–62)	50 (44–58)	0.4763	52 (43–62)	55 (43–62)	0.6194
Weight (kg)						
Median (interquartile range)	69 (60–75)	70 (67–85)	0.4227	77 (63–85)	66 (57–74)	0.0280 *
BMI						
Median (interquartile range)	25.51 (23.09–28.20)	29.33 (25.19–31.93)	0.1391	28.26 (23.23–30.84)	25.55 (20.98–27.51)	0.0323 *
BMI	25 (92.6%)	2 (7.4%)		5 (20.8%)	19 (79.2%)	
Underweight or Healthy body weight			2.08 [0.40–9.20]			0.51 [0.15–1.67]
Overweight or Obese (classes 1–3)	30 (85.7%)	5 (14.3%)	0.4039	13 (34.2%)	25 (65.8%)	0.2626
Tumor highest diameter (mm)	20.5 (15–30)	18.5 (12–27.5)	1	25 (20–35)	20 (13.5–28.7)	0.0889
ypT						
0	33 (91.7%)	3 (8.3%)	2.00 [0.41–9.82]	10 (28.6%)	25 (71.4%)	0.95 (0.31–2.87)
1 or 2	22 (84.6%)	4 (15.4%)	0.3932	8 (29.6%)	19 (70.4%)	0.9275
ypN						
Negative	52 (91.2%)	5 (8.8%)	6.93 [0.93–51.79]	18 (31.6%)	39 (68.4%)	5.15 [0.27–98.7]
Positive	3 (60%)	2 (40%)	0.0591 ^^^	-	5 (100%)	0.2756
Biological subtype of cancer						1.09 (0.32–3.68)
A, B1, B2, HER2+	39 (88.6%)	5 (11.4%)	0.97 (0.17–5.55)	13 (29.5%)	31 (70.5%)	0.8893
TN	16 (88.9%)	2 (11.1%)	0.9772	5 (27.8%)	13 (72.2%)	
Biological subtype of cancer						2.68 (0.30–24.05)
A, B1, B2, TN	49 (89.1%)	6 (10.9%)	1.36 (0.14–13.31)	17 (30.9%)	38 (69,1%)	0.3775
HER2+	6 (85.7%)	1 (14.3%)	0.791	1 (14.3%)	6 (85.7%)	
Response to NAC						
No response	16 (84.2%)	3 (15.8%)	1.83 (0.37–9.11)	8 (30.8%)	18 (69.2%)	1.16 (0.38–3.50)
Response	39 (90.7%)	4 (9.3%)	0.4616	10 (27.8%)	26 (72.2%)	0.798

RI—radioisotope; SPIO—superparamagnetic iron oxide; OR—odds ratio; *p ^a^*—chi-square test result; *p*
^*b*^—odds ratio test result; SLNs—sentinel lymph nodes; BMI—body mass index; ypT—post neoadjuvant therapy pathological T stage; ypN—post neoadjuvant therapy pathological N stage; A—luminal A; B1—luminal B human epithelial receptor-negative; B2—luminal B human epithelial receptor-positive; HER2+—human epithelial receptor-positive; TN—triple-negative; NAC—neoadjuvant chemotherapy *—statistically significant result; ^—a trend into statistically significant result.

## Data Availability

The data presented in this study are available on request from the corresponding author.

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
