# Peer review of "A Propensity Score Matched Analysis of Superparamagnetic Iron Oxide versus Radioisotope Sentinel Node Biopsy in Breast Cancer Patients after Neoadjuvant Chemotherapy"

_cancers, 2022, doi:10.3390/cancers14030676_

Round 1

Reviewer 1 Report

This is a very interesting study using SPIO for SNB after NAC.

All cases were performed Ax dissection and have no therapeutic disadvantage, but SNB for T3-4 breast cancer in generally contraversial.

In this paper Staging after NAC is described. If cN1 cases which are diagnosed before NAC is converted to ycN0 after NAC, SNB is not indicated generally.

Although the negative conversion from cN1 to ycN0 has a high false negative rate, it has been proved in tha large study that the false negative rate can be reduced by removing 4 or more LN2.

Reviewer 2 Report

I read the manuscript with interest. Breast diseases represent a large part of my clinical practice. The topic is of great interest, where new methods of detecting the sentinel lymph node have become research focus. The authors tried to demonstrate the usefulness of a new method of perparamagnetic iron oxide (SPIO).
My main comments are as follows:

  1. 1. Expanding the introductory part, Unfortunately the broader discussion of axillary study is missing, I would advise the authors to revisit their literature search and at least add these works:
    Cirocchi et al. New classifications of axillary lymph nodes and their anatomical-clinical correlations in breast surgery. World J Surg Oncol. 2021 Mar 29; 19 (1): 93. doi: 10.1186 / s12957-021-02209-2. Available online
    De Luca A et al. Evaluation of the Effectiveness of a Synthetic Glue and a Fibrin-Based Sealant for the Prevention of Seroma Following Axillary Dissection in Breast Cancer Patients. Front Oncol. 2020 Jul 17; 10: 1061. doi: 10.3389 / fonc.2020.01061. Available online. So that this part can be strengthened. Add some points on the role of indocyanine green on sentinel lymph node detection.

2. Please clarify the paragraph from line 56 to line 60

3. Both the objectives and the conclusions of this review are broad and radical

4. The images are of good quality and standardized.

The effort by the authors is considerable, the study is well designed to demonstrate and provide useful information to readers.

Accept after Minor Revisions
